# Improving the Efficacy of Botulinum Toxin for Cervical Dystonia: A Scoping Review

**DOI:** 10.3390/toxins15060391

**Published:** 2023-06-09

**Authors:** Roberto Erro, Marina Picillo, Maria Teresa Pellecchia, Paolo Barone

**Affiliations:** Department of Medicine, Surgery and Dentistry “Scuola Medica Salernitana”, Neuroscience Section, University of Salerno, Via Allende 43, 84081 Baronissi, SA, Italypbarone@unisa.it (P.B.)

**Keywords:** accuracy, ultrasound, electromyography, guided injections, daxi

## Abstract

Cervical dstonia (CD) is a chronic disorder with a significant detrimental impact on quality of life, requiring long-term treatment. Intramuscular injections of botulinum neurotoxin (BoNT) every 12 to 16 weeks have become the first-line option for CD. Despite the remarkable efficacy of BoNT as a treatment for CD, a significantly high proportion of patients report poor outcomes and discontinue the treatment. The reasons that drive sub-optimal response or treatment failure in a proportion of patients include but are not limited to inappropriate muscle targets and/or BoNT dosing, improper method of injections, subjective feeling of inefficacy, and the formation of neutralizing antibodies against the neurotoxin. The current review aims to complement published research focusing on the identification of the factors that might explain the failure of BoNT treatment in CD, highlighting possible solutions to improve its outcomes. Thus, the use of the new phenomenological classification of cervical dystonia known as COL-CAP might improve the identification of the muscle targets, but more sensitive information might come from the use of kinematic or scintigraphic techniques and the use of electromyographic or ultrasound guidance might ensure the accuracy of the injections. Suggestions are made for the development of a patient-centered model for the management of cervical dystonia and to emphasize that unmet needs in the field are to increase awareness about the non-motor spectrum of CD, which might influence the perception of the efficacy from BoNT injections, and the development of dedicated rehabilitation programs for CD that might enhance its effectiveness.

## 1. Introduction

Cervical Dystonia (CD) is a movement disorder characterised by involuntary contractions leading to abnormal head movements, postures, or both. Additional clinical features of CD are neck pain and head tremors, which might be in some cases the predominant manifestation and might lead patients to seek medical advice. CD is the commonest form of idiopathic adult-onset dystonia, with an estimated prevalence of 3 to 28/100,000 people in Western countries [1,2]. CD is a chronic disorder, despite spontaneous remissions having been described in a minority of cases [3], with a significant detrimental impact on patients’ quality of life [4]. As such, it requires long-term treatment and repetitive intramuscular injections of botulinum neurotoxin (BoNT) every 12 to 16 weeks have become the first-line option for CD, with both randomized clinical trials (RCTs) and open studies demonstrating its high effectiveness and safety profiles, even on the long term [5]. BoNT blocks acetylcholine release at the neuromuscular junctions, therefore inhibiting muscle contractions [6]. Currently, there are four BoNT/A products and one BoNT/B product that are available for dystonia (Table 1) but new formulations might be approved soon. Although BoNT is regarded as the first-line option for CD, between 17.9 and 46% of patients report inefficacy and discontinue the treatment [7,8,9,10,11,12,13]. The reasons for discontinuation are not entirely understood [14]. Patients disclosing no or suboptimal response to BoNT have been classically stratified in primary and secondary non-responders, based on whether a satisfactory outcome was never observed or was lost over repeated cycles of injections. This partition reflects the assumption that the reasons for inefficacy might be different between these two groups of patients and, consequently, available options to restore efficacy [14,15,16]. However, with the exception of the formation of neutralizing antibodies (NABs) in patients who have been exposed for a long time to BoNT, it seems that reasons for inefficacy might be shared between these two groups and include, but are not limited to, inappropriate muscle targets and/or BoNT dosing, improper method of injections and subjective feeling of inefficacy. It is important to note that these factors not only drive treatment failure in a proportion of patients but also determine sub-optimal responses in other patients. In line with this, time series analyses on an individual level demonstrate that only 40% of CD patients display the expected U-shaped curve of BoNT efficacy across a single treatment cycle [17]. These patients had longer BoNT injection intervals, showed a better match to objective symptom assessments, and were characterized by a stronger certainty to control their somatic symptoms (e.g., internal medical locus of control) [17]. This highlights how patient perspectives and expectations from treatment represent a crucial factor explaining, at least partially, the efficacy of BoNT therapy [16,18] and opens the question of how to best measure the outcomes after BoNT injections.

Whereas few articles have been published focusing on the identification of the factors explaining the failure of BoNT [15,16,17], the current scoping review aims to complement these works in highlighting possible solutions to improve the efficacy of BoNT therapy in CD. 

To this aim a literature search via Pubmed was performed using the combination of “cervical dystonia” [term A] with “botulinum toxin” [term B] AND “efficacy”, “failure”, “ultrasound”, “electromyography”, “technique”, or “antibodies” [term C]. One author (RE) reviewed the titles and, whenever appropriate, abstracts of the retrieved articles. The reference lists of relevant articles were also checked to eventually include reports missed through the electronic search. The final reference list was generated based on relevance to the topics covered in this review.

## 2. Correct Identification of Muscle Targets in CD

One of the commonest reasons which might explain BoNT therapy failure would be incorrect identification of the clinical pattern of CD. Suboptimal/wrong muscle targeting has been reported to account for about 37% of cases with unsatisfactory responses following BoNT treatment for CD in one study [16], this being the second commonest reason after inadequate BoNT dosing explaining BoNT treatment failure. Of note, up to 75% of cases in this series achieved satisfactory responses after revision of the original treatment plan [16], which highlights the presence of correctible factors in most of these patients. Importantly, this study also suggested that specific CD subtypes, including the anterocollic CD, tremor-dominant CD, and CD with refractory neck pain, might be more challenging to treat [16].

Classically, CD had been classified into four types depending on the main vector implicated in the abnormal posture (e.g., torticollis, laterocollis, retrocollis, and anterocollis) with the torticollis representing more than 50% of cases [19,20]. However, following the proposal for a new phenomenological classification of dystonia known as the COL-CAP concept [21], it has been shown that only about 16% of patients with CD have “pure” forms (i.e., with only one implicated vector) [22] and that up to 6 COL-CAP subtypes could be identified in a single patient [23].

The COL-CAP concept set out by Reichel proposes to differentiate between CD subtypes based on whether the dystonic activity involves muscles that act between the skull and C2 vertebra or between the vertebrae C2 and C7, C2 being regarded as a kind of fixed point (Figure 1). For instance, when muscles that induce a rotation rostral to C3 are dystonic, the head shows a pivotal movement in relation to the neck, which is the torticaput subtype of CD. Conversely, if the movement takes place caudal to C2 a rotation of the neck in relation to the trunk is observed, determining the torticollis subtype of CD. This crucial distinction applies to all three dimensions of movements (Figure 1) and would help in the selection of target muscles (Table 2). Although this represents a captivating concept, evidence demonstrating better outcomes following its application in clinical practice are very scarce [23,24] and, instead, it seems that the most commonly injected muscles have remained unchanged over the past few decades despite this new proposal [25].

In addition to careful clinical evaluation, it is hypothesised that the use of an objective technique would improve muscle selection and, in turn, treatment outcome. Thus, a few studies assessing the role of polymyographic electromyography (pEMG) in CD patients with poor response to BoNT treatment, including one randomized clinical trial, have shown that more than half of these patients can achieve a satisfactory response after the injections without the need to increase BoNT doses [26,27,28,29,30,31]. Although in most CD patients a good clinical response can be achieved with clinical evaluation alone [32], one might speculate that treatment outcomes can be further improved with pEMG. However, it has been also suggested that pEMG might lack specificity. In a study comparing frequency analysis of EMG recordings and pEMG, the former showed very high specificity (98%) in identifying dystonic muscles by the detection of a “dystonic” drive of about 4–7 Hz, but poor sensitivity (17%) [33]. Given that only 40 of the 280 possible muscle pairs were considered non-dystonic based on pEMG, the authors hypothesized the low sensitivity of frequency analysis was due to poor specificity of pEMG, which might not be able to discriminate between dystonic muscles and compensatory active muscles [33]. Alternative EMG techniques, including EMG coherence, spectral analysis, and muscle activation patterns during isometric contractions [34,35,36,37], have also been suggested to be helpful in the correct identification of dystonic muscles. It should be noted, however, that all these techniques made use of surface EMG electrodes, thus limiting their applicability to superficial neck muscles. 

One study has shown that kinematic technology may improve treatment outcomes by guiding physicians to better tailor muscle selection and BoNT dosing [38]. Thus, a group of 14 CD patients underwent an assessment with motion sensors to detect the angular deviation of multiaxial neck/head posturing during both static and dynamic conditions and, dependent on the extent and direction of the primary CD pattern as evaluated from the kinematic graphs, muscles associated with each plane of motion were targeted for treatment [38]. The results were then compared to those obtained in a group of 14 CD patients who received injections based on visual assessment alone. Injections guided by kinematic analysis of CD biomechanics resulted in faster optimal muscle selections, lower BoNT-A doses, and better outcomes in terms of dystonia severity and associated disability, as compared to visual determination alone [38], suggesting this technique might be implemented in clinical practice because of its non-invasiveness, despite being time-consuming. 

Finally, a body of research has been produced that positron emission tomography (PET) or single photon emission computed tomography (SPECT) imaging can be useful in the identification of dystonic muscles. Three studies investigated 18-FDG-PET imaging as a method of identification of dystonic muscles for BoNT injections in CD patients by using the maximum standardized uptake value as a criterion for selecting a target [39,40,41]. Baseline disability scores and numbers of hypermetabolic (and therefore injected) muscles, including deep cervical muscles, were observed to be significant predictors of good outcomes [41]. Despite the good results following the injections, none of the studies used a control group to compare the outcomes, nor is there clarity about the reasons explaining the poor response in a subset of patients [39,40,41]. 

Chen and colleagues instead evaluated the usefulness of [^99m^Tc]MIBI SPECT for the identification of dystonic muscles using as gold standard clinically identified muscular targets in patients with CD who were good responders to BoNT [42]. The muscle/background ratio of targeted muscles was significantly higher than normal muscles and the sensitivity and specificity of this technique in correctly identifying dystonic muscles were demonstrated to be 93.2% and 88.5%, respectively [42]. In a subsequent study, it was shown that severity reduction of the dystonia severity was significantly higher at 3- to 6-month post-injections in patients in whom injections were guided by [^99m^Tc]MIBI SPECT than in those in whom target muscles were selected by clinical evaluation alone [43]. Moreover, the re-injection interval was longer in the former than in the latter group [43]. These results were confirmed in a subsequent double-blind, randomized study [44], suggesting this technique might become a useful tool to aid the identification of targets for BoNT injections.

## 3. Correct Injections–US/EMG

The accuracy of the injections is fundamental for the effectiveness and safety of BoNT treatment [45]. Several methods can be used for BoNT injections, ranging from the simple anatomical guidance, which is based on the palpation of anatomical landmarks, to different assisted methods which make use of electrophysiology (i.e., EMG-guided) or imaging [ultrasound (US) or cranial tomography (CT)] techniques. Quite intuitively, the accuracy of injections is expected to be superior using these techniques than with anatomical guidance alone [46]. Indeed, a study supported the role of EMG indicating that injections performed using only anatomic landmarks are quite unreliable, the sternocleidomastoid muscle being correctly reached in 83% of cases and the levator scapulae muscle in only 47% of cases [47]. Moreover, the EMG-guidance has been further demonstrated to produce greater effectiveness of BoNT injections [26] and a prolonged benefit with lower incidence of side effects [48] when compared to anatomically guided injections. More sophisticated electrophysiology techniques such as EMG interference pattern [49] and high-density surface EMG to localize the motor endplate [50], have been preliminarily suggested to further increase the accuracy of injections and, in turn, be helpful in lowering the BoNT dose improving the overall safety of BoNT treatment.

Similarly, Kreisler and colleagues used the US to locate the tip of the needle and the BoNT pool after the needle had been inserted [51]. Thus, analysing over 50 CD patients and 300 injections it was demonstrated that the overall accuracy of needle placement by anatomical guidance was about 75%, with the splenius capitis representing the worst targeted muscle [51]. These results broadly mirrored those of a cadaveric study comparing US-guided and anatomically guided injections into cervical muscles, which demonstrated an accuracy of about 95–100% and of about 55–80%, respectively [52]. Although these findings would support the concept the inaccurate delivery of the BoNT might account for suboptimal or negative outcomes, only two studies have formally tested this hypothesis [53,54]. Thus, by comparing US-guided injections and injections guided by identification of anatomical landmarks, it was shown that Toronto Western Spasmodic Torticollis Rating Scale (TWSTRS) disability and pain subscales significantly decreased only in the former group and that the total TWSTRS and its severity subscale, as well as measures of quality of life, showed a greater reduction with US than in the group assigned to anatomically guided injections [53]. On the other hand, another study failed to demonstrate the significant difference in any outcomes between the US-guided and non-guided injections group [54]. It should be noted, however, that in the latter study there was a recruitment bias in the way that patients assigned to the US guidance were previously considered non-responders: therefore, the suggestion was put forward that US guidance made it possible to obtain the same results in the most severe (or the most demanding) patients as in the best responders, who received non-guided injections [54].

Preliminary evidence has been also produced with CT-guided injections, which might be preferred over US guidance in some patients because of a poor sonic window to target such muscles as the longus collis (LC) [39,55]. Thus, one study demonstrated the feasibility of this approach which further resulted into the improvement of clinical symptoms as assessed with the TWSTRS and Tsui scales [39]. An alternative approach to target the LC in the anterocollis subtype of CD is fluoroscopy, which serves to ensure that the needle does not pass into the vicinity of the vertebral artery. A preliminary study using this approach demonstrated its feasibility and relatively good outcomes in 8 out of 10 injections [56].

## 4. Phenotype and Progression of CD and Their Relationship with BoNT Efficacy

A rather overlooked factor that might explain the partial or no efficacy of BoNT injections might be related to the phenotype and natural history and CD itself. Some, if not most, of the clinical features that have been demonstrated to influence patients’ satisfaction with BoNT treatments would be “non-modifiable”. Nonetheless, these are important to recognize because they can be exploited to correctly counsel patients, as continued below. 

Whereas it is commonly accepted that some CD subtypes such as antecollis and antecaput might be more resistant to BoNT treatment [57], mainly because the involved muscle to target are deeply located and there is a higher chance of treatment-emergent side effects (i.e., dysphagia), there are somehow conflicting findings on whether some other clinical features related to the phenotype of CD might or not predict a positive response to BoNT. For instance, it has been reported that the majority of patients with minimal severity of CD when beginning BoNT therapy did not experience significant improvement, for at least the first 3 to 4 injection cycles [58]. Conversely, another study found the opposite in the way that higher severity and related disability of CD predicted a poor outcome [59]. Similar inconsistencies can be found in terms of cervical pain, with some reporting it as a good prognostic factor in terms of treatment outcome [60] and others arguing the opposite [59], as well as in terms of head tremor [61,62]. Additionally, worse treatment satisfaction correlated with shorter time intervals between injection cycles [59], but it is unclear whether this could be a proxy of higher CD severity. Finally, young age at first treatment [63,64] and shorter time between disease onset and treatment initiation [62,65] have been further suggested to positively influence the outcome.

Regarding the natural history of CD, it is a common perception that most patients show a steady progression of their focal dystonia to reach the maximal disability after about 5 years [65], after which a further progression of the disease might be exhibited in terms of dystonia spread to other body regions [66]. However, when looking at patterns of long-term injections, some studies reported the total BoNT dose to be stable or to be even reduced [67,68], supporting the aforementioned conception, whereas in others the BoNT dose per session was found to significantly increase over time [69,70,71], which might hint at a progression of the disease beyond the initial 5 years from onset. Indeed, more recent studies have suggested that the degree of complexity of CD may increase with disease duration even under ongoing treatment with BoNT [62,72]. In one study, when patients were stratified into a subgroup of patients with a good effect of BoNT therapy and a second subgroup of patients with an unsatisfactory effect, those with an increase of the pattern of the complexity of CD (i.e., head deviation observed in all three planes of motion) were more frequently found in this second subgroup of subjects reporting an unsatisfactory effect [72]. Another study demonstrated that new symptoms including pain, head tremor and/or jerks, and loss of voluntary head control might develop and expand the clinical spectrum of CD patients under BoNT therapy and, therefore, drive the perception of partial benefit from the injection despite of improvement of head position [62]. Similarly, another study found that, compared to CD patients with moderate to good treatment satisfaction, those with none or low BoNT efficacy had an increased incidence of cervical pain and had higher coexistence of oromandibular dystonia [59]. 

The picture delineated above clearly indicates that: (1) we lack robust and solid data about the clinical features of CD that might predict the outcomes following repeated injections in BoNT-naïve patients; and (2) that CD should be conceived as a progressive disorder with a substantial proportion of patients developing new symptoms that were not present at BoNT therapy initiation or showing spread of dystonic signs to additional body regions. This further calls for the need of collecting data from longitudinal studies which might be used for correctly counselling patients and helping them to set appropriate expectations from BoNT treatment. 

## 5. Additional Factors: BoNT Dosing, Injections Intervals, and Neutralizing Antibodies

Quite intuitively, the efficacy of BoNT injections depends also on the correct dosing of the neurotoxin of each target muscle. Indeed, in one study, the most common reason for non-responsiveness was inadequate dose, as demonstrated by the possibility to achieve a satisfactory outcome by increasing the dose after referral of these patients to a specialized centre for BoNT injections [16]. However, it should be noted that 42.8% of these subjects required a BoNT amount exceeding the maximum dose recommended in the product package insert to reach satisfactory responses [16]. These results were only partly confirmed by another study which showed that, when patients were stratified according to the received dose, rates of satisfaction were slightly higher when BoNT was dosed per recommended dose ranges than treatment at the extremes of dosing [73]. Although it was not clearly stated, this might have happened because of the development of side effects in a proportion of patients in the high-range dose [73]. This would be in line with the findings of another study showing how the development of side effects represents an independent risk factor for partial/non-responsiveness to BoNT injections [15]. Therefore, dosing must be tailored to the single patients according to CD phenotype and severity, but evidence-based recommendations for dose ranges are available per single target muscle for all BoNT-A products and should be, in general terms, respected [18].

Additionally, studies have shown that satisfaction with symptom control is highest at peak symptomatic effect and lowest at the end of the cycle when the therapeutic response has waned [74,75], with the impact of CD on quality of life following the same ‘rollercoaster’ pattern in most patients [76], even those undergoing regular injections and reporting reasonable efficacy at peak effect. This highlights that symptom re-emergence is common and has a significant impact on patients’ satisfaction with the treatment, with some patients reporting early waning of treatment benefit well before the typical 12-week reinjection interval. Therefore, on the one hand, greater awareness of the therapeutic profile of BoNT-A products should lead to better informed therapeutic discussions and planning. On the other hand, these findings open the question of what would be the best injection interval to improve treatment outcomes. This is particularly relevant if one further considers that time-series analyses to detect individual responsiveness to BoNT treatment demonstrate that only about 40% of patients with CD display the expected inversed U-shaped curve of BoNT efficacy across a single treatment cycle [17]. As mentioned above, CD patients who followed the expected outcome course typically tolerate longer BoNT injection intervals [17], highlighting how the re-injection schedule can be customized according to patients’ responses. Interestingly, an RCT comparing a patient-initiated treatment model to the usual care in subjects with blepharospasm or hemifacial spasm, revealed no significant differences in terms of treatment efficacy but highlighted the potential of the new model to save healthcare costs and reduce anxiety [77]. Moreover, patients using this new model were equally satisfied in the service and confident in their care as those receiving treatment as usual [77]. In support of this notion, a post hoc analysis of the Interest in CD-2 study found that patients who attended clinics that allowed some flexibility in injection cycles (to meet individual patient needs) had slightly longer abo-BoNT/A injection intervals than those who attended clinics with fixed schedules, suggesting that many patients treated flexibly are able to go longer than the standard interval [78]. However, the opposite, a phase 4, open-label, randomized, non-inferiority study has been recently performed to compare two inco-BoNT/A injection intervals (Short Flex: 8 ± 2 weeks; Long Flex: 14 ± 2 weeks) in patients with inadequate benefit from standard injection intervals [79]. The results showed that injection cycles < 10 weeks for inco-BoNT/A are effective, non-inferior, and tend to have numerically more favorable clinical outcomes, including a subjective satisfaction score, than longer intervals for treating CD patients with early waning of clinical benefit [79]. Moreover, shorter injection intervals did not increase adverse events or lead to loss of treatment effect, at least as observed over a period of eight injection cycles [79]. It should be noted, however, that this study suffered from multiple protocol amendments [79], so that immunogenicity data were too sparse to allow for meaningful conclusions, and one concern, as discussed in more detail below, regards the possibility of development of NABs with shorter intervals. Moreover, this study only demonstrated non-inferiority, rather than superiority, of the short interval schedule in comparison with the usual one, and pharmacoeconomic considerations might apply if one further considers the time to retreatment observed in studies using different BoNT products (i.e., mean injection intervals for treatment cycle 1 of ~12 weeks in the German inco-BoNT/A study [80], of ~14 weeks with ona-BoNT/A in the CD-PROBE study [20], and of ~16 weeks with abo-BoNT/A [81]).

Repeated injections can lead to the development of NABs against BoNT and therefore reduced responsiveness to therapy with partial or complete treatment failure. As mentioned above, one of the factors that has been implicated in the development of NABs against BoNT is a short interval between injection cycles. In fact, an early study analysing about 560 CD patients treated over a period of 8 years found that those developing treatment failure had more frequent injections per year and more “booster injections” in between treatment cycles, compared to non-resistant patients [82]. There are, however, additional factors contributing to the development of NABs against BoNT. It has been suggested that a higher content of bacterial proteins play a role [83] and a potential adjuvant activity of the complexing proteins is also discussed [84,85]. For instance, NABs had been detected in more than 17% of CD patients following ona-BoNT/A treatment [82,86] before the protein content of this product was modified in 1998, following which NABs were only reported in 1.2% of the patients receiving ona-BoNT/A [87,88]. For abo-BoNT/A, a NABs rate of about 2% has been reported [68]. It should be noted, however, that the reported rates of NABs grossly vary according to different studies and ranges, in more recent surveys on long-term treated patients, from 0.6% [67] to >10% [89,90]. One of the reasons explaining such discrepancies might stand in the method used to detect NABs as well as their interpretation. One study comparing three different methods for NABs detection (i.e., the mouse lethality assay, the mouse diaphragm assay, and the sternocleidomastoid test) did not demonstrate the superiority of any of them and further showed that neither correlated with the subjective complaint of therapy failure [91]. Therefore, although there is evidence of a non-linear increase of NABs formation with treatment duration [89,90,92], the current opinion [93,94] would be that their formation occurs only in a very small percentage of patients and does not explain the large proportion of cases with unsatisfactory outcomes. In fact, a meta-analysis including not only patients with dystonia demonstrated that more than half of the so-called “non-responders” do not have NABs [95]. It is advised, however, to test patients with secondary and permanent treatment failure with one of the available methods and, in case of results consistent with the presence of NABs, to switch to alternative BoNT/A products, preferring formulations with low anticigenity [18,96,97]or to rima-BoNT/B [98,99]. 

## 6. Final Considerations and Conclusions

Despite the remarkable efficacy of BoNT as a treatment for CD, it remains a challenge for the treating clinicians and there has been increasing awareness that: (1) patients with CD tend to rate their outcomes less enthusiastically than their physicians [20,72,100]; and (2) that a considerable proportion of CD patients discontinue the treatment [14]. Several factors, which have been the subject of this scoping review, have been implicated in the lack of or partial response to BoNT. Addressing these factors in clinical practice has the potential to improve treatment outcomes (Table 3). 

Quite obviously the identification of the dystonia pattern and target muscles is a fundamental prerequisite to obtaining good outcomes from the injections. However, following the COL-CAP concept, which poses that different treatment protocols can be pursued depending on the caput/collis phenotype, only two retrospective studies have been performed demonstrating better outcomes [23,24]. Future research should therefore focus on this aspect. On the other hand, however, the COL-CAP classification, does not consider the presence of head tremor, which is common in CD and among the main factors that might explain the subjective reporting of failure after the injections. In this regard, it should be acknowledged that none of the existing rating scales for CD has been tested for responsiveness after BoNT. Moreover, they do not capture these novel aspects of the disorder including the COL-CAP concept [101]. This opens the question of how to measure the efficacy of BoNT with initial suggestions advocating the use of objective markers, at least to measure changes in the motor component of the disorder [102]. At the other end of the spectrum, however, one should consider that non-motor symptoms such as pain and mood dysfunction are integral to CD and might significantly influence the subjective perception of treatment efficacy [16]. These aspects should be considered when counselling BoNT-naïve patients to help them set correct expectations from the treatment. This applies also to the re-injections schedule planning as the duration of the efficacy does not always correspond to the usual 3–4 month period adopted for the injection cycles. While a certain degree of flexibility in scheduling can be offered without impacting health system resources [67,68], it seems wise to develop new long-lasting BoNT formulations that might help in this regard. Accordingly, a novel BoNT/A product (i.e., daxi-BoNT/A) has been recently formulated with a proprietary stabilizing excipient peptide (RTP004), which has been shown to enhance binding of the neurotoxin to neuronal surfaces and therefore to enhance the likelihood of neurotoxin internalization [103]. In the ASPEN-1 study, daxi-BoNT/A at 125 U and 250 U significantly improved TWSTRS total scores, with a median duration of efficacy of 24 and 20 weeks, respectively, which compares very favourably with the 12–14 week duration reported for approved BoNT/A products [104]. daxi-BoNT/A was generally safe and well tolerated, with dysphagia reported in only 1.6% and 3.8% of CD patients treated with 125 U and 250 U [105], respectively, which is considerably lower than the incidence of dysphagia reported in registration trials of other BoNT/A formulations. A long-term, open-label, efficacy and safety study (ASPEN-OLS) has been conducted, with over 350 CD patients being enrolled and being eligible for up to four doses of daxi_BoNT/A over 48 weeks [106]. The study was completed in May 2021, but data were not available at the time of writing. It seems, however, that this new formulation holds promise in the treatment of CD as it might mitigate some of the patients’ complaints regarding the re-emergence of symptoms after a short period of time from the injections. Similarly, pre-clinical studies have suggested that the use of BoNT products combined with polysaccharides such as globular chitosan significantly prolongs the effective duration of the toxin [107]. It is therefore expected that additional long-lasting formulations will be tested for CD to address this issue. 

The clinical practice standards involve the identification of muscle targets by clinical examination alone, but it remains to be established whether it should be considered the best approach to adopt, even applying the COL-CAP classification. On the other hand, the only, relatively accessible, technique to aid the identification of the target muscles in CD is the pEMG [26,27], but there is insufficient evidence to recommend it in all CD patients, also in view of its relative invasiveness requiring at least eight needle electrodes to be concomitantly placed in different neck muscles. Nonetheless, pEMG remains the only technique which can be used in clinical practice to improve treatment efficacy in those patients reporting a poor response to BoNT injections, while awaiting the validation of a less invasive technique for the identification of dystonic muscles, which might come from the use of wearable sensors [38]. Once the target muscles have been identified, it is necessary to ensure that needle placement is accurate. While in the field of spasticity, the use of any guide (i.e., US. EMG, and electrostimulation) for BoNT delivery has been demonstrated to be superior to the palpatory technique (level A of evidence) [108], this is not the case for CD [53,54] and future, well-conducted studies should be performed in this population. In addition to the above, the use of US might further allow to target deeply located muscles such as the Obliquus capitis inferior and semispinalis cervicis, which might be frequently involved in CD according to the COL-CAP concept (Table 2). Moreover, preferential injection sites within the selected muscles have been suggested to aid the chemical neurolysis [50,109,110].

In case of documented poor clinical response, one should take into account disease progression and review muscle selection and BoNT doses as well as consider immunoresistance. Beyond the case that has been discussed above about NABs in patients who have been previously treated with BoNT injections, one should also consider the rare occurrence of pre-existing NABs due to, for instance, botulism exposure [111] or to previous vaccinations against BoNT [112]. A pentavalent vaccination (against serotypes A–E) was used among the US military personnel during the Gulf War XXX [112]. BoNT/A titers, were detectable in about 30% of soldiers who had received a vaccination 1–2 years prior to the testing and in up to 99% of those who received a subsequent booster dose [112].

Addressing these aspects will likely improve the efficacy of BoNT treatment in CD. However, in these concluding remarks, we would like to emphasize that additional unmet needs are increasing awareness about the non-motor spectrum of CD [113], which might influence the perception of the efficacy of BoNT injections, and the development of dedicated rehabilitation programs for CD that might enhance the efficacy of the treatment.

## Figures and Tables

**Figure 1 toxins-15-00391-f001:**
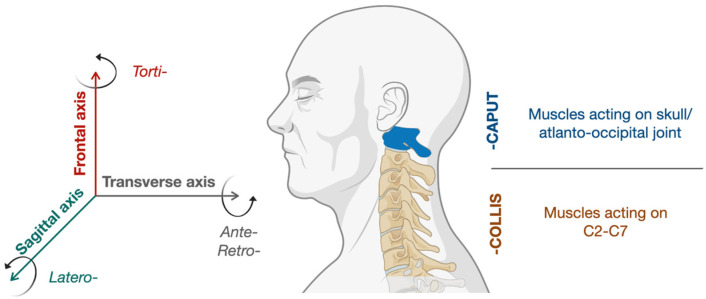
Anatomical basis of COL-CAP concept for cervical dystonia. Two levels of movement around the three axes with C2 as a fixed point determine the subtypes of cervical dystonia.

**Table 1 toxins-15-00391-t001:** Summary of product characteristics of Botulinum toxin formulations approved or tested for dystonia.

	Ona-BoNT/A	Abo-BoNT/A	Inco-BoNT/A	Neu-BoNT/A	Rima-BoNT/B	Daxi-BoNT/A
Molecular weight (kDa)	900	500–900	150	900	700	150
Strain	Hall	Hall	Hall	Hall	Bean	Hall
Presence of accessory proteins	Yes	Yes	No	Yes	Yes	No
Presence of HSA	Yes; 500 mcg	Yes; 125 mcg	Yes, 1 mg	Yes; 1 mg	500 µg/mL	No
Excipients	Sodium chloride	Lactose	Sucrose	Sodium chloride	Sodium chloride, Sodium succinate	PS20, sugar, buffer, excipient peptide (RTP004)
Stabilization	Vacuum drying	Lyophilization	Lyophilization	Freeze drying	Solution	Lyophilization
Purification method	Crystallization	Chromatography	NA	NA	Chromatography	Chromatography
Shelf-life once reconstituted (h)	36	24	36	NA	Few hours	72
Can be stored at room temperature unreconstituted	No	No	Yes	No	No	Yes

HSA human serum albumin, NA not applicable, PS20 polysorbate-20.

**Table 2 toxins-15-00391-t002:** Muscles involved in the subtypes of cervical dystonia according to the COL-CAP concept.

	Ipsilateral Muscles	Contralateral Muscles	Bilateral Muscles
**Torticaput**	Obliquus Capitis Inferior *Longissimus Capitis* *Splenius capitis*	Trapezius (pars descendens) Sternocleidomastoideus *Semispinalis Capitis* (*pars medialis*)	
**Torticollis**	Semispinalis Cervicis Levator Scapulae *Splenius Cervicis* *Longissimus Cervicis*		
**Laterocaput**	Sternocleidomastoideus Trapezius (pars descendens) Splenius Capitis *Semispinalis Capitis* *Longissimus Capitis* *Levator Scapulae*		
**Laterocollis**	Levator Scapulae Semispinalis Cervicis *Scalenus Medius* *Longissimus Cervicis*		
**Anterocaput**			Longus Capitis Levator Scapulae *Sternocleidomastoideus*
**Anterocollis**			Scalenus Medius Levator Scapulae *Longus Collis*
**Retrocaput**			Obliquus Capitis Inferior Semispinalis Capitis Trapezius (pars descendens) *Splenius Capitis*
**Retrocollis**			Semispinalis Cervicis

Secondary muscle targets are indicated in italics. Additional subtypes represent the combination of those indicated in the table. For instance, the “lateral shift” subtype is the combination of laterocollis to one side and laterocaput to the opposite side. Similarly, the “sagittal shift” phenotype results from the combination of anterocollis and retrocaput.

**Table 3 toxins-15-00391-t003:** Main issues for lack of or partial efficacy of BoNT for Cervical Dystonia with suggested solutions and future directions.

Issues	Current Solutions	Future Directions
Incorrect identification of muscle targets	- Application of the COL-CAP concept - Use of polymyographic electromyography	Use of kinematic sensors Use of scintigraphic techniques (e.g., [^99m^Tc]MIBI SPECT)
Accuracy of BoNT injections	- Use of an EMG/US guidance	
Inappropriate BoNT dosing	- Use of recommended dose ranges	
Short-lasting effect of BoNT dosing	- Patient-centered model for scheduling re-injection sessions (even before the canonical 3 month period)	Use of long-lasting BoNT products Dedicatd rehabiliation programs to prolong BoNT efficacy
Neutralizing antibodies	- Switch to alternative BoNT/A products, preferring formulations with low anticigenity or to rima-BoNT/B	
Expectations	- Correct counselling - Addressing non-motor symptoms	Patient-centered model for the “holistic” management of CD
Other (disease progression/head tremor/difficult phenotype)	- Evaluate patient over the disease course and tailor injections accordingly	

SPECT: Single Photon Emission Computed Tomography; EMG: Electromyography; US: Ultrasound; CD: Cervical Dystonia.

## Data Availability

Not applicable.

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
