# Peer review of "Improving the Efficacy of Botulinum Toxin for Cervical Dystonia: A Scoping Review"

_toxins, 2023, doi:10.3390/toxins15060391_

Round 1
Reviewer 1 Report
To make the information easier, I recommend that a table that summarizes all the data should be included, for each main idea, as the narrative style is sometimes hard to follow.
Also, the conclusions should be more synthetic.
Author Response
We thank the Reviewer for the positive evaluation and suggestions. We have added an accompanying table to the discussion to summarize in a bullet-point fashion the identified issues and the possible solutions.
Reviewer 2 Report
The subject of the study is interesting, but the conduction of the methodoly used in the scoping review was not appropriated.
The abstract needs to be improved, the methodology is not properly described, the results and discussion need to be improved and the references need to be up-to-dated. Thus, the manuscript needs to be needs to be rewritten to be consider to review.
The English Language needs to be improved.
Author Response
We respectfully disagree with this Reviewer's opinion, which is in contrast to the other two Reviewers who positively evaluated our Review. Given that the criticisms were rather unspecific we were not able to address punctually the Reviewer's concerns, nor understand in which way the references were not updated wince we included all relevant references including those published in 2021-2022. Nonetheless, we have detailed the methodology applied to obtain the references used in the manuscript. The "results" section has been structured according to the main identified issues and obviously follows a narrative style. We would be happy to punctually address the Reviewer's concerns, if explicitly provided, including the inclusion of any missed references.
Reviewer 3 Report
The review article well describes the etiology and pathogenesis of all variants of cervical dystonia and diagnostic methods . All variants of modern treatment of this pathology with the use of botulinum toxin are exhaustively described. The known problems and difficulties characteristic of different variants of CD and the possibilities of overcoming them are discussed. It is important that the article mentions promising treatment methods based on prolonged forms of botulinum toxin, as well as the influence of the immune response on the effectiveness of therapy.
It makes sense to add information about prolonged forms of botulinum toxin based on the use of polysaccharides (e.g., chitosan). And also discuss the effect of pre-existing antibodies not directly related to prior botulinum toxin injections. Otherwise, the article is ready and can be accepted for publication.
Author Response
We thank the Reviewer for the positive evaluation and useful suggestions. We have added a comment in the concluding paragraph about both the prolonged formulations of BoNT based on the use of globular chitosan and the rare occurrence of pre-existing NABs against BoNT.
Reviewer 4 Report
The article discusses additional factors in the discussion and that can affect the efficacy of botulinum toxin (BoNT) injections for the treatment of cervical dystonia (CD).
BoNT Dosing: The effectiveness of BoNT injections depends on the correct dosing of the neurotoxin for each target muscle. Inadequate dosing can lead to non-responsiveness. Some patients may require doses exceeding the recommended maximum to achieve satisfactory results. However, higher doses can increase the risk of side effects.
Injection Intervals: Patient satisfaction with symptom control is highest at the peak symptomatic effect of BoNT and lowest at the end of the treatment cycle when the therapeutic response has waned. CD patients often experience symptom re-emergence and may report early waning of treatment benefits before the typical 12-week reinjection interval. Customizing the reinjection schedule based on individual patient response can lead to longer injection intervals and improved outcomes. Shorter injection intervals (< 10 weeks) for certain BoNT-A formulations have shown effectiveness without increasing adverse events or compromising treatment effects.
Neutralizing Antibodies (NABs): Repeated BoNT injections can lead to the development of NABs, which reduce the responsiveness to therapy and may cause treatment failure. Factors contributing to NAB development include a short interval between injection cycles, higher content of bacterial proteins, and potential adjuvant activity of complexing proteins. The rates of NAB formation vary among studies, and the method used for NAB detection can influence results. While NAB formation increases with treatment duration, it occurs in only a small percentage of patients and does not explain all cases of unsatisfactory outcomes. Non-responders can exist even without the presence of NABs. Testing patients with secondary or permanent treatment failure for NABs is advised, and switching to alternative BoNT-A products with low antigenicity or rima-BoNT/B may be considered.
Another factor that should be considered is injection point, along with the dosing. Please regard neuromuscular distribution articles “Anatomical guide for botulinum neurotoxin injection: Application to cosmetic shoulder contouring, pain syndromes, and cervical dystonia” and “Effective botulinum toxin injection guide for treatment of cervical dystonia”.
Overall, the review seems to have great ideas with full of information in treating cervical dystonia.
Major revision should be conducted.
Author Response
We thank the Reviewer for the positive evaluation. We have included the two suggested references and commented this point on as follows:
“Moreover, preferential injection sites within the selected muscles have been suggested to aid the chemical neurolysis [51,113,114].”